

# An autonomous cloud detection algorithm using single ground-based infrared radiometer for the Tibetan Plateau

Linjun Pan[1], Yinan Wang[1], Yongheng Bi[1]

[1]Laboratory of Middle Atmosphere and Global Environment Observation, Institute of Atmospheric

Physics, Beijing,100029, China

*Correspondence to:* Yinan Wang (wangyinan@mail.iap.ac.cn)

**Abstract.** Accurate cloud detection over the Tibetan Plateau (TP) is crucial for understanding regional weather patterns and global climate dynamics. Yet, it remains challenging due to harsh environmental conditions and sparse observations. While

ground-based infrared radiometers offer a promising solution through downwelling infrared brightness temperature (IRBT) measurements, existing algorithms require supplementary meteorological data often unavailable in remote TP regions.    This study presents a novel cloud detection algorithm that operates solely on IRBT data from a single ground-based infrared radiometer, addressing the critical need for autonomous

cloud monitoring in resource-limited environments. The algorithm integrates complementary spectral and temporal analysis approaches: the spectral test identifies cloud presence by comparing observed IRBT against statistically derived clear-sky diurnal cycles, and the temporal test detects clouds through IRBT variability analysis using sliding standard deviation calculations. A key innovation includes a

normalization procedure that effectively mitigates dust contamination effects—a persistent challenge in the arid TP environment that can introduce errors exceeding 40°C. Validation against 13 months of radiosonde data demonstrates robust



performance with agreement rates exceeding 70% in most months, with particularly effective performance during the wet season. This work provides a practical and cost-

effective solution for autonomous cloud monitoring over the TP, with potential for application in other regions with limited observational data.

## 1    Introduction

The Tibetan Plateau (TP) plays a critical role in shaping weather and climate patterns on regional and global scales (Wu and Zhang, 1998; Duan and Wu, 2005; Wu

and Chou et al., 2013; Zhang et al., 2018; Ge et al., 2019; Fu et al, 2020; Wu et al., 2024). These effects are closely linked to clouds, which influence convection systems and radiative forcing over the TP (Duan and Wu, 2006; Ma et al., 2021; Bo et al., 2016; Bao et al. 2019; Wu et al. 2024). The related heating can even modulate East Asian summer monsoon (EASM), such as the onset, duration, and total precipitation of the

rainy season in South China (the first stage of the EASM) (Duan et al., 2020). Detecting and understanding cloud properties is therefore essential for studying cloud impacts on the TP's climate .

Cloud observations provide valuable information not only for assessing atmospheric processes but also for improving retrieval accuracy in satellite and ground-

based remote sensing. Even clouds with small optical depths can significantly affect the accuracy of retrieved vertical profiles of temperature and humidity, precipitable water vapor, and liquid water (Hewison, 2007; Turner, 2007; Cadeddu and Turner, 2011). However, the TP's harsh climate and complex terrain result in sparse and unevenly distributed observation stations, making it challenging to monitor clouds



effectively. As a result, there is a need for simple and cost-effective cloud observation methods.

Ground-based infrared radiometers, which are small, portable, and relatively inexpensive, have been widely deployed for cloud detection. These instruments measure downwelling infrared radiance in the form of infrared brightness temperature

(IRBT) within the atmospheric window band. On cloudy days, the detected radiance includes contributions from clouds, the atmosphere between the cloud and the radiometer, and the atmosphere above the cloud. Clouds in the troposphere significantly increase IRBT compared with clear-sky conditions, enabling cloud detection by comparing IRBT values (Brocard et al., 2010; Ahn et al., 2015).

However, this approach has limitations. Strong inversion layers or the presence of thick aerosols and haze can cause clear skies to be misclassified as cloudy (Sutter et al., 2004; Ahn et al., 2015). Conversely, optically thin cirrus clouds often produce IRBT increments that are too weak to reliably distinguish from clear-sky conditions. Brocard et al. (2010) addressed this limitation by using detrended fluctuation analysis to

estimate the temporal variability of IRBT for detecting cirrus clouds. Ahn et al. (2015) developed a cloud detection algorithm that combines spectral and temporal tests on IRBT data, using auxiliary data such as historical radiosonde observations and real-time surface atmospheric parameters. While effective, these approaches rely on additional instruments, making them less practical in regions like the TP, where

supplementary observations are often unavailable.

This study addresses the need for a cloud detection algorithm that relies solely on IRBT measurements from a ground-based infrared radiometer. The proposed algorithm integrates spectral and temporal tests to identify cloud presence. The spectral test compares observed IRBT with the statistical diurnal cycle of infrared radiance, while the temporal test assesses IRBT variability against a statistical threshold. This method eliminates the dependence on auxiliary data, making it more suitable for remote and resource-limited regions like the TP.

The rest of this paper is organized as follows: Section 2 describes the dataset and instrumentation. Section 3 details the new cloud detection algorithm. Section 4 presents the evaluation results, and Section 5 summarizes the main findings and discusses the uncertainties and future directions.

## 2    Instrumentation and Data

### 2.1    Infrared Brightness Temperature (IRBT) Data

This study utilizes data from an infrared radiometer (model KT19.85), manufactured by Heitronics. The radiometer, mounted atop a ground-based microwave radiometer, functions as an auxiliary instrument for capturing downwelling infrared radiation. It has a spectral range of 9.4-11.8 μm, with peak sensitivity concentrated between 10 and 11.5 μm.

The radiometer was deployed on a rooftop platform at Tibet University's Najin campus, located in Lhasa, Tibet, at 3,650 meters above sea level. Continuous measurements of IRBT began in June 2021, and a total of 13 months of data have been collected.



## 2.2 Radiosonde Data

Radiosondes are launched in Lhasa twice daily at 12:00 and 24:00 UTC. Unless otherwise stated, all times in this paper follow UTC. The radiosonde launch site is approximately 4 kilometers from the Najin campus. The corresponding radiosonde data are used to validate the cloud detection algorithm.

## 3 Methodology

The cloud detection algorithm integrates two complementary approaches: a spectral test and a temporal test, combining their strengths to achieve optimal cloud detection. The spectral test evaluates the absolute IRBT values against the statistical clear-sky IRBT diurnal cycle, while the temporal test examines the variability of IRBT over time. Before the algorithm application, quality control procedures are implemented to ensure data reliability by mitigating the effects of noise and environmental contamination.

### 3.1 Normalization of IRBT

The infrared radiometer used in this study captures total downwelling infrared radiation via a vertically positioned reflective lens. A transparent protective cover surrounds the lens to mitigate contamination from the weather like rain, snow, and dust. Despite this, field observations on the Tibetan Plateau are inevitably contaminated by dust accumulation, a consequence of the arid and windy conditions in this region. This contamination results in a gradual increase in IRBT values, potentially obscuring cloud-related signals and degrading detection accuracy.

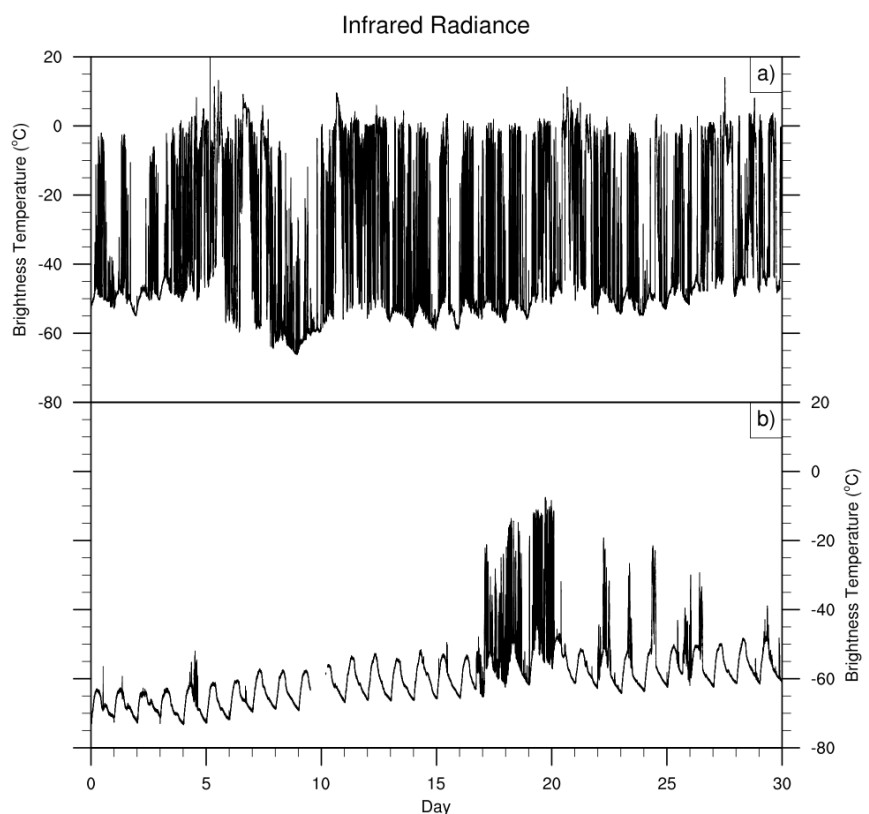

Figure 1: The observations of IRBT in June (Fig. a) and November (Fig. b) 2021.

Figures 1 and 2 illustrate examples of IRBT observations. In June and November 2021, the daily minimum IRBT shows a gradual increase over time, which is associated with dust accumulation. The increase can be interrupted by abrupt drops caused by aperiodic manual lens cleaning events (Fig. 1a and Fig. 2). In some extreme situations, IRBT noise introduced by dust accumulations can exceed the infrared signals from clouds. In March 2022, a sharp IRBT decrease of ~40°C on the 9th day after the removal of accumulated dust (Fig. 2). These findings underscore the critical need to correct for dust-induced noise.



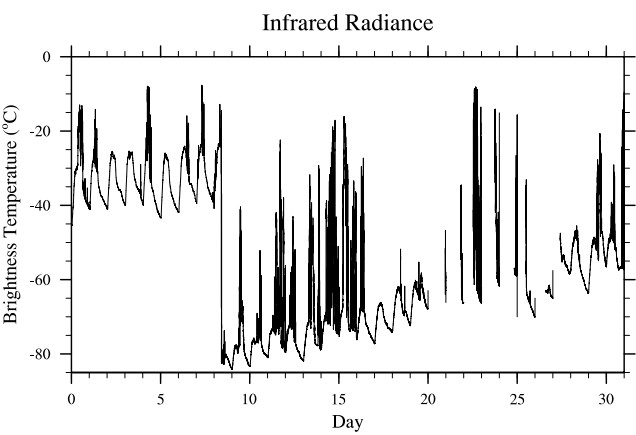

Figure 2: The observations of IRBT in March 2022.

To address this, we developed a normalization method that effectively mitigates the impact of dust accumulation on the cloud detection algorithm. The steps are as follows:

1. Daily Segmentation: Separate the observation data into daily intervals.

2. Minimum Extraction: Identify the daily minimum IRBT value.

3. Data Normalization: Subtract the daily minimum IRBT from the corresponding daily dataset.

The normalization approach can significantly reduce dust-induced noise, providing reliable data for subsequent analysis. In Fig. 3, we show the normalized IRBT in June and November 2021 as an example. While normalization improves data quality, it may introduce errors under specific weather conditions, which will be discussed in the following section.





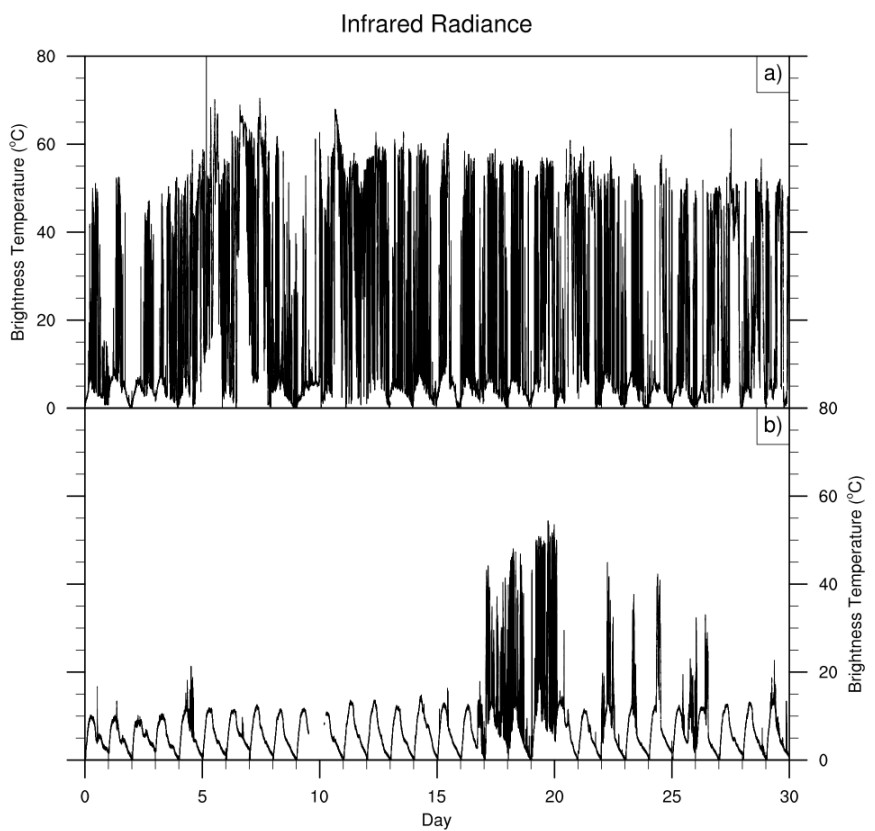

Figure 3: The normalized infrared brightness temperatures in June (Fig. a) and

November (Fig. b).

## 3.2 Spectral Test

The spectral test utilizes the clear-sky IRBT diurnal cycle to detect cloud presence.

140 During the dry season (e.g., November), distinct diurnal cycles of the IRBT are

observed (Fig. 1b). During the wet season (e.g., June), the sky is frequently cloud-

covered, resulting in higher IRBT values than those during the dry season, but the

diurnal cycles can still be observed (Fig. 1a). The amplitude of the IRBT diurnal cycles

during the dry seasons is stronger than those during the wet seasons, primarily driven





145     by former stronger temperature diurnal cycles on clear days. This seasonal difference

in IRBT diurnal cycles becomes more obvious after the normalization (Fig. 3).

On clear days, IRBT closely follows the diurnal temperature cycle, which remains

relatively stable over short periods (e.g., several days). On cloudy days, observed IRBT

includes additional contributions from clouds, deviating from the typical clear-sky

150     diurnal pattern. Thus, detecting deviations from the clear-sky IRBT diurnal cycle is

central to the cloud detection algorithm. Calculating the clear-sky IRBT diurnal cycle

is the key component in the spectral test.

To address this, we developed a method for extracting the clear-sky IRBT diurnal

cycle solely from infrared radiometer observations, without supplementary data. The

155     method accounts for varying seasonal conditions, adjusting the temporal resolution to

20 minutes during the wet season and 10 minutes during other months. The steps are as

follows:

1.     Data Segmentation: For each target time point in the diurnal cycle, extract a

corresponding sequence of IRBT observations from adjacent days. During the wet

season, the extraction window is doubled to include more clear-sky data due to

prolonged cloud cover.

2.     Low-Value Selection: From the extracted sequence, select the lowest 5% of

IRBT values and calculate their average as the diurnal cycle value for the target time.

3.     Sliding Calculation: Repeat the above process for each time point to construct

the complete diurnal cycle.



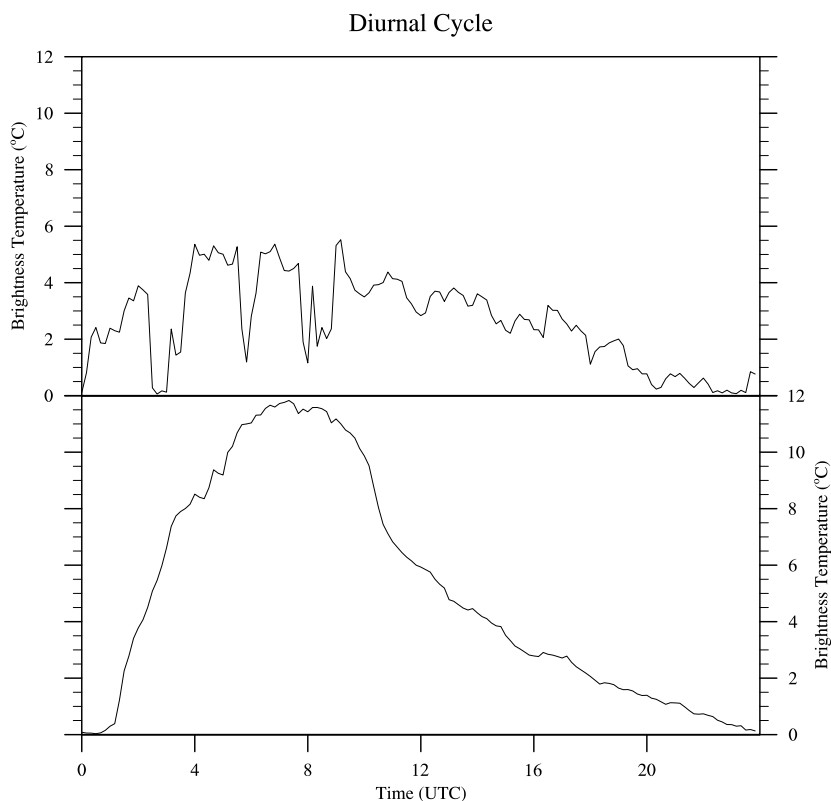

Figure 4: The calculated clear-sky IRBT diurnal cycle on June 15 and November 15, 2021.

Figure 4 illustrates the extracted clear-sky IRBT diurnal cycles for June 15 (wet season) and November 15 (dry season). While both exhibit trends consistent with diurnal temperature variations, the winter cycle (November) shows a larger amplitude due to greater daily temperature differences. The summer cycle (June) exhibits fluctuations, likely due to residual cloud signals and potential errors from the

normalization process. When the cloud covers throughout the day, the daily minimum IRBT occurs at any time. This may cause a low value in the calculated diurnal cycle.



To mitigate these issues, we use the maximum IRBT of the clear-sky diurnal cycle as the reference for the spectral test. This ensures robustness against false signals caused by cloud contamination or normalization errors.

180        Steps of the Spectral Test:

1. Derive the clear-sky IRBT diurnal cycle using the method described above.

2. Extract the maximum IRBT from the diurnal cycle.

3. Detect cloud presence by comparing observed IRBT with the maximum IRBT.

If the observed IRBT exceeds 150% of the maximum, the sky is classified as cloudy;

otherwise, it is clear.

To account for special conditions such as fog, the threshold for cloud detection is set conservatively above the maximum IRBT of the clear-sky diurnal cycle. The larger threshold inevitably introduces errors in judgment. It may misclassify clouds with low optical depths at night as clear skies. This limitation is addressed by incorporating an

additional detection method described in the next section.

### 3.3   Temporal Test

On cloudy days, variations in cloud properties, such as optical depth and cloud base height, introduce significant changes in ground-based infrared radiance. In contrast, clear-sky conditions result in relatively smooth IRBT curves. Figure 5a

illustrates the normalized IRBT in November 2021, where clear-sky periods exhibit smooth fluctuations, while cloudy periods display pronounced variability.




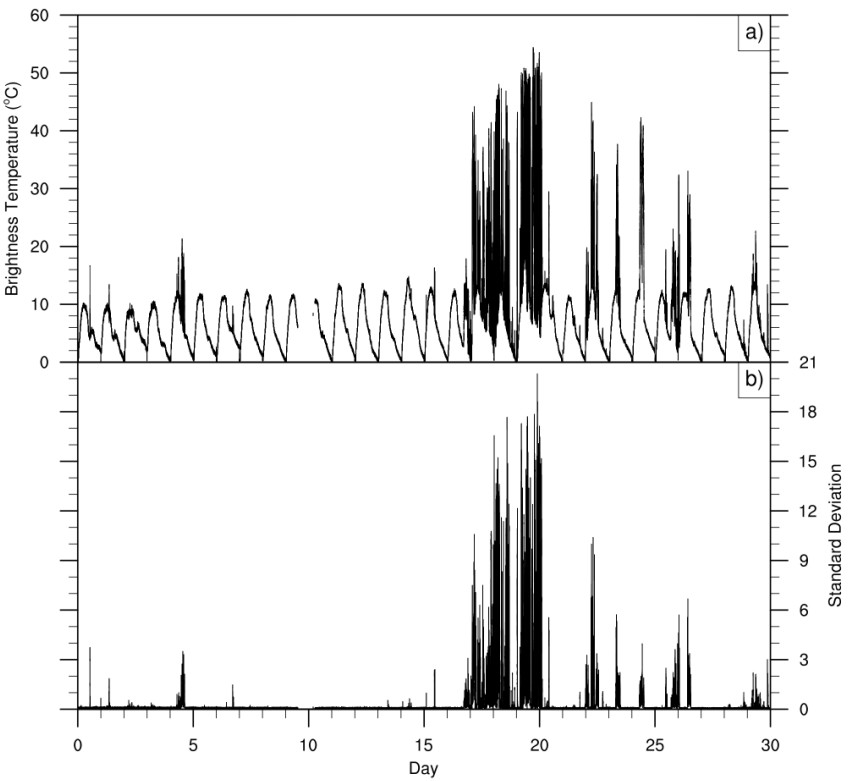

Figure 5: The normalized infrared brightness temperatures (Fig. a) and the
corresponding calculated standard deviations (Fig. b) in November 2021.


Based on these characteristics, a temporal variability-based method is developed
to detect clouds. This approach uses the standard deviation of normalized IRBT as the
criterion for cloud presence. When the standard deviation exceeds a specific threshold,
the sky is classified as cloudy.

Figure 5b shows the sliding standard deviation of IRBT in November 2021, which
aligns closely with the corresponding IRBT variations. For a more detailed
demonstration, Figure 6 presents normalized IRBT data for November 23, 2021 (Fig.



6a), along with the corresponding standard deviations (Fig. 6b). Under clear-sky

conditions, the IRBT standard deviations remain around 0.1, whereas cloudy conditions

exhibit significantly higher values, typically exceeding 0.3. Based on long-term

analysis, a threshold of 0.3 is established for cloud detection.

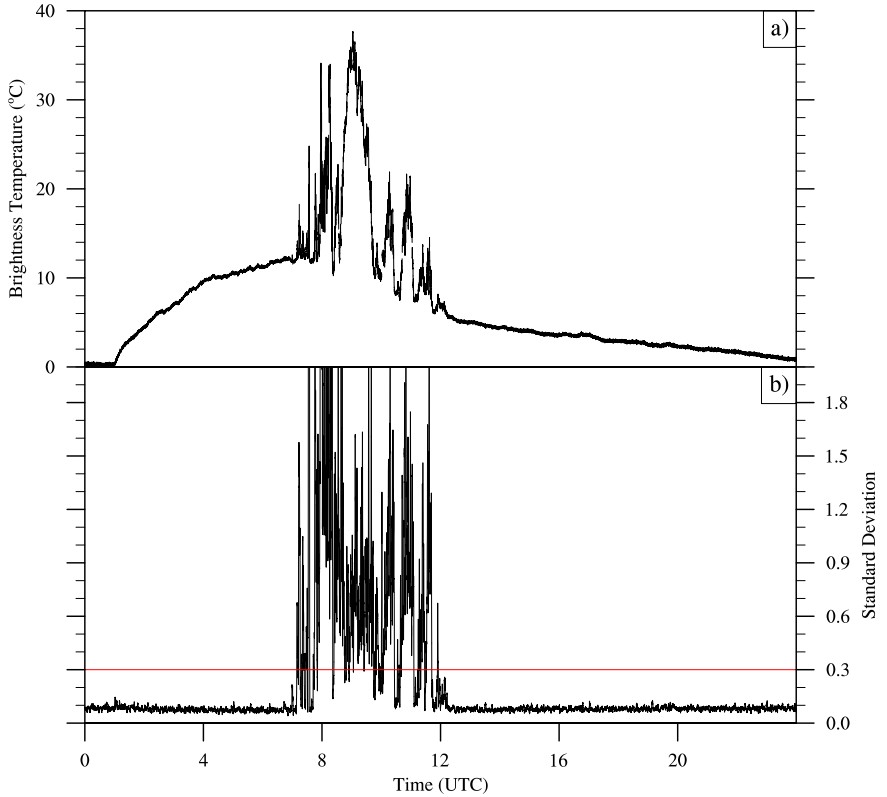

Figure 6: The normalized infrared brightness temperatures (Fig. a) and the
corresponding calculated standard deviations (Fig. b) on November 23, 2021.


Steps of the Temporal Test:

1.   Standard Deviation Calculation: The standard deviation of normalized IRBT

is calculated over a sliding one-minute window. The temporal resolution of IRBT is



two seconds. Each window contains 30 data points, ensuring sufficient statistical

confidence.

2.   Cloud Detection: If the calculated standard deviation exceeds 0.3, the sky is

classified as cloudy; otherwise, it is classified as clear.

The temporal test is particularly effective in detecting clouds with small optical

depths but significant temporal variability—scenarios that may be missed in the

spectral test. However, this method may misclassify stable clouds with minimal

variation, such as stratus clouds, as clear skies. Combining the temporal and spectral

tests enhances the robustness of the overall cloud detection algorithm.

### 3.4   Overall Flowchart of the Cloud Detection Algorithm

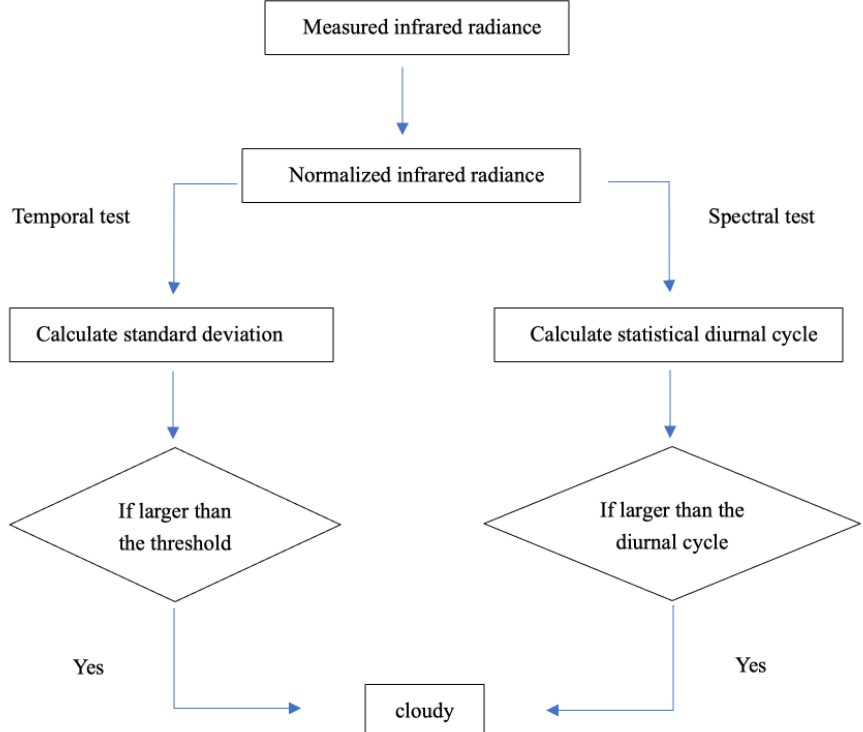

230                            Figure 7: Flow chart of the cloud detection algorithm.



The integrated cloud detection algorithm combines the spectral and temporal tests described in the above two sections. The workflow is summarized in Figure 7 and involves the following steps:

1. Normalization of IRBT: Process the observation data to obtain normalized

IRBT values, reducing the contamination from dust accumulation.

2. Spectral Test: Calculate the clear-sky IRBT diurnal cycle and extract its maximum value. If the observed IRBT exceeds 150% of the maximum, the sky is classified as cloudy; otherwise, it is classified as clear.

3. Temporal Test: Calculate the sliding standard deviation of the normalized

IRBT over a one-minute window. If the standard deviation exceeds 0.3, the sky is classified as cloudy; otherwise, it is classified as clear.

4. Final Judgment: Combine the results of the spectral and temporal tests. The sky is classified as clear only if both tests indicate clear-sky conditions. Otherwise, it is classified as cloudy.

Figure 8 illustrates the application of the algorithm using observational data on November 24, 2021. The observed IRBT (Fig. 8a), normalized IRBT (Fig. 8b), corresponding standard deviations (Fig. 8c), and clear-sky IRBT diurnal cycle (Fig. 8d), and are shown. The final detection results are presented in Figure 9b, where clear-sky and cloudy conditions are represented by blue and red lines, respectively. The results

align well with IRBT variations, accurately capturing detailed changes in sky conditions, such as those occurring around 8:00 AM. This demonstrates the algorithm's effectiveness in identifying cloud presence under varying conditions.





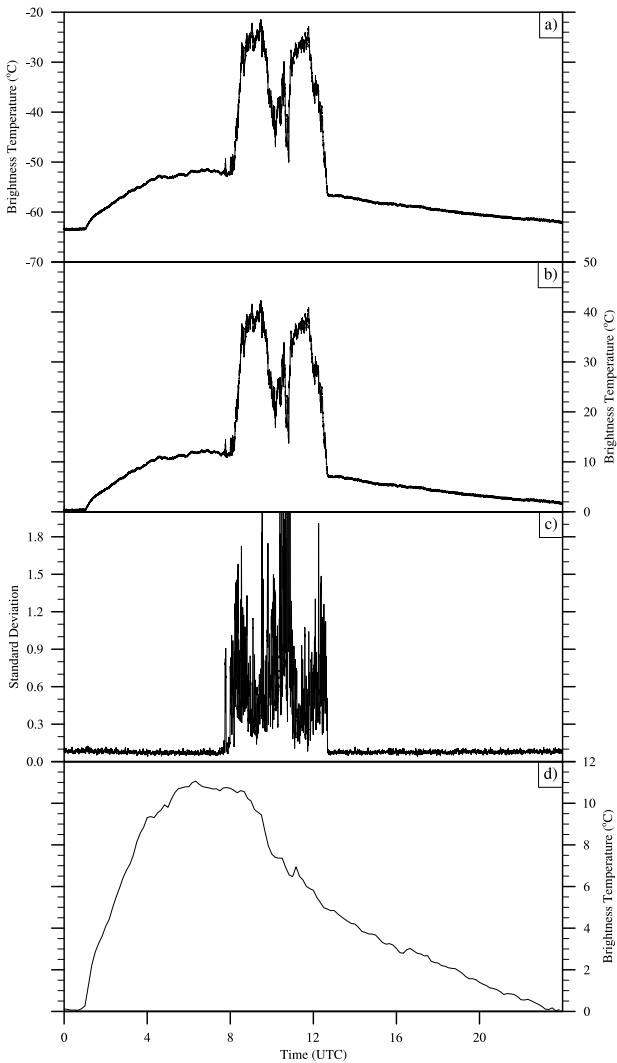

Figure 8: The observations of IRBT (Fig. a), the normalized infrared brightness

temperatures (Fig. b), standard deviation (Fig. c), and clear-sky IRBT diurnal cycle

(Fig. d) on November 24, 2021.


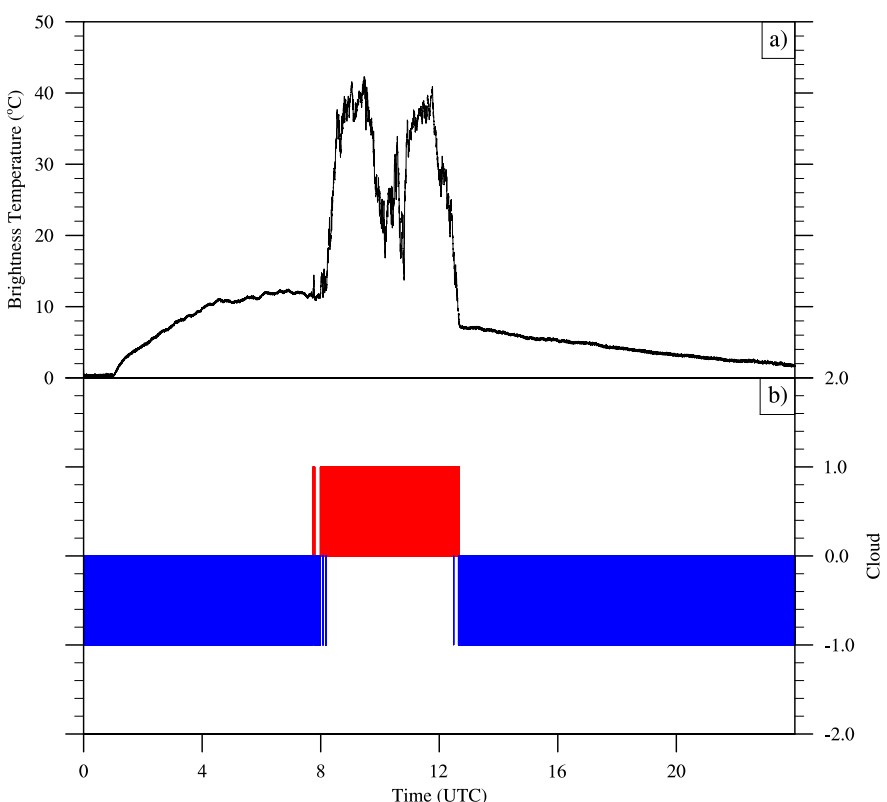

Figure 9. The normalized infrared brightness temperatures (Fig. a) and the

corresponding cloud results (Fig. b) on November 24, 2021. Plus one (red line) and

minus one (blue line) represent cloudy and clear-sky respectively in Fig. b.

## 4    Validation of the cloud detection algorithm

Since no other relevant meteorological instruments are deployed in Tibet

University, the accuracy of the cloud detection algorithm is assessed by comparing with

the radiosonde data provided by the Lhasa Meteorological Bureau. The radiosonde



launch site, located 4 kilometers from the infrared radiometer, conducts measurements twice daily at 00:00 and 12:00 UTC.

The high-resolution vertical profile of relative humidity from radiosonde can be used to detect the presence and vertical structure of clouds (Wang et al., 1999; Cai et al., 2014; Li et al., 2021). If the corresponding relative humidity is larger than a certain threshold, it is determined that there are clouds at that height (Wang et al., 1995; Zhou and Qu, 2010). The thresholds were set be larger than 80% in previous studies (Zhang et al., 2010; Zhou and Qu, 2010; Cai et al., 2014; Li et al., 2021). Due to the different climates, the threshold for cloud detection in different seasons and regions should be selected differently (Zhang et al., 2010). Considering that the climate in Lhasa is generally drier, the threshold for cloud detection in this paper is chosen as 80% in wet seasons from June to August, and 70% for the other times.

In addition to the selection of threshold value, the following three aspects are also considered in the comparison:

1) The main contribution to ground-based infrared radiation comes from the troposphere. Meanwhile, clouds are mainly concentrated in the troposphere. On the other hand, the sonde balloon drifts with the wind in the rising process, which may be far away from the radiometer. Therefore, the relative humidity below 10 KM is selected for the comparison.

2) The sonde balloon needs some time to acquire the atmosphere information. Therefore, the corresponding infrared observation time for the comparison is selected 20 minutes around the sonde time.




3) Considering the short-term noise and other impacts, the infrared cloud judgment

should keep a certain time in the comparison. If the judgment cannot satisfy the time,

the sky is judged to be clear for infrared cloud detection. The temporal resolution of the

infrared radiometer is two seconds. In the comparison, the cloud judgments must

exceed 30 times during the sonde time, which means the cloud exists for more than a

minute.

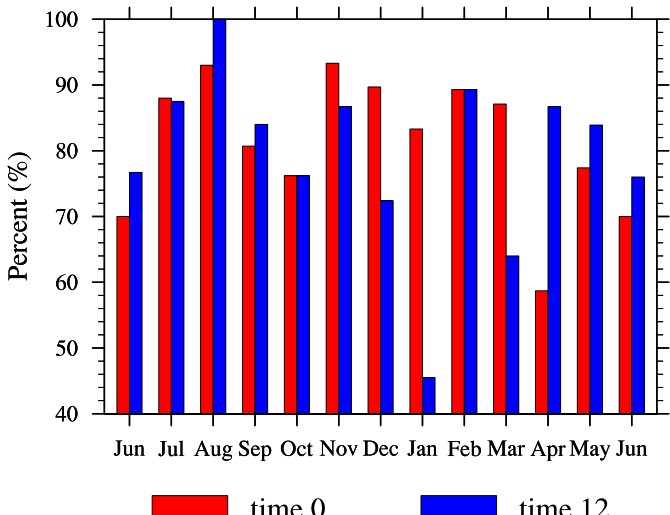

Figure 10: The comparison results between infrared radiometer and radiosonde. Red
and blue boxes represent the radiosonde launch times 00:00 and 12:00 (UTC). The
agreement percentage reflects the proportion of consistent cloud detection results
between the two methods relative to total valid observations each month.

Fig. 10 shows the comparison results over 13 months (June 2021 – June 2022).

The red and blue boxes represent radiosonde launch times at 00:00 and 12:00 UTC,

respectively. The agreement percentage reflects the proportion of consistent cloud



detection results between the two methods relative to total valid observations each

month.

Overall, the algorithm performs well, achieving agreement rates above 70% in

most months. Some winter times show slightly reduced performance, with agreement

rates around 50%. An analysis of inconsistent results reveals the following primary

causes:

1.   Spatial Inconsistency: The infrared radiometer observes radiation from the

zenith, detecting only clouds within its field of view. Radiosondes, which drift with the

wind during ascent, may observe atmospheric profiles outside the radiometer's

coverage. Additionally, the 4 km distance between the radiometer and the radiosonde

launch site increases the likelihood of discrepancies, particularly for clouds with small

horizontal scales.

2.   Temporal Inconsistency: The exact radiosonde launch time can vary, and

balloon ascent takes time. Rapid cloud dynamics may lead to misalignment between

the radiometer and radiosonde observations.

3.   Threshold Errors in Radiosonde Detection: No statistical analyses of cloud

detection thresholds for relative humidity profiles have been conducted in Lhasa.

Seasonal and weather-dependent variations in threshold values could contribute to

misjudgments in radiosonde-based cloud detection.

Despite these limitations, the algorithm demonstrates robust performance,

particularly during the wet season, underscoring its effectiveness for cloud detection on

the Tibetan Plateau.



## 5  Conclusions and Discussions

This study develops an integrated cloud detection algorithm designed for the ground-based infrared radiometer, specifically addressing the challenges of observations on the Tibetan Plateau. The algorithm combines spectral and temporal

tests, leveraging their complementary strengths to improve cloud detection accuracy. It is notable for relying solely on infrared radiometer data, eliminating the need for supplementary observational instruments—a significant advancement over previous methods.

A critical contribution of this work is the development of a normalization method

to address contamination caused by dust accumulation on the radiometer lens. This issue, exacerbated by the arid and windy climate of the Tibetan Plateau, has been largely overlooked in previous studies. By normalizing the IRBT data on a daily basis, this method effectively mitigates the impact of dust contamination, ensuring reliable inputs for the algorithm.

The spectral test identifies cloud presence by analyzing deviations from the clear-sky IRBT diurnal cycle, while the temporal test evaluates the variability of IRBT over short time scales. Although each test has own limitations, such as misclassifying clouds with thin optical depths or small variability, their integration provides a robust framework for cloud detection. This integration allows the algorithm to overcome

individual shortcomings, offering improved performance in capturing cloud signals under diverse atmospheric conditions.



Validation of the algorithm was conducted using radiosonde data from the Lhasa Meteorological Bureau over 13 months. Agreement rates exceeded 70% in most months, demonstrating the algorithm's effectiveness, particularly during the wet season.

However, reduced wintertime performance highlights challenges related to spatial and temporal inconsistencies. For instance, the drift of radiosonde balloons and the fixed zenithal field of view of the radiometer may lead to discrepancies, particularly for clouds with small horizontal scales. Furthermore, the lack of statistical analyses for cloud detection thresholds in Lhasa introduces potential errors in the radiosonde-based

comparisons. These limitations underscore the need for improved validation strategies.

Looking ahead, a new dual millimeter-wavelength cloud radar and infrared imaging detector (MWII), supported by the Ground-based Space Environment Monitoring Network project, has been deployed at the Yangbajing Whole Neutral Atmospheric Observing Station offers an opportunity for significant advancements.

The MWII, which combines active radar with a passive infrared radiometer, provides detailed information on cloud vertical structure and optical properties. This system will allow for more comprehensive evaluations and refinements of the integrated algorithm, ensuring greater accuracy and applicability across diverse atmospheric conditions.

*Data availability*. The data shown in the paper is available on request from corresponding author.

*Author contributions*. LP and YW led the paper writing, LP and YB made the measurements, LP and YW made the calculations, and all co-authors participated on the writing and commenting the manuscript.


*Financial support*. This research was supported by the Strategic Priority Research Program of Chinese Academy of Sciences (Grant No. XDA0470Y0Z), the Basic Scientific Research Project of Institute of Atmospheric Physics during the 14th Five-year Plan period and we acknowledge the use of data from the Chinese Meridian Project.

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
