# Peer review of "An autonomous cloud detection algorithm using single ground-based infrared radiometer for the Tibetan Plateau"

_EGUsphere, 2025_

## Author Comment (AC1)

Dear Reviewer:

We sincerely thank the reviewer for the valuable and constructive comments on our manuscript "An autonomous cloud detection algorithm using single ground-based infrared radiometer for the Tibetan Plateau" (ID: EGUSPHERE-2025-2876). We have carefully revised the manuscript accordingly. Below, we provide point-by-point responses (reviewer's comments in italic, our responses in normal font).

*Q1. Recent advancements of cloud detection in the Tibetan Plateau regions can be considered in the Introduction.*

**Response:** Thank you for your suggestion. The recent advancements in cloud detection have been added to the introduction. "Since the 1980s, meteorological satellites such as CloudSat, CALIPSO, Himawari-8, and FengYun-4A have been used to reveal cloud characteristics over the Tibetan Plateau (Yan et al., 2019; Yi, 2019; Wang et al., 2020; Liu et al.,2021). In contrast to these top-down observations, ground-based instruments—including cloud radars, lidars, and all-sky imagers—provide essential bottom-up data for validating satellite products and investigating local cloud properties in greater detail (Song et al., 2017; Huo et al., 2021; Luo et al., 2024; Zhao et al., 2024)." (Line 44-50, Section 1, Page 3)

*Q2. To enhance clarity, authors may consider presenting a comprehensive flowchart that displays the primary algorithm's logic and main steps. While Figure 7 illustrates the algorithm flow, it would be beneficial to include additional details described in the text, such as the steps of normalization and the calculation of the clear-sky IRBT diurnal cycle. This comprehensive visualization would assist readers in understanding the entire algorithm. In addition, Figure 7 can be repositioned to appear earlier in the methodology section.*

**Response:** According to your suggestion, we revised the flow chart, which involves more details in normalization, calculation and threshold used (Fig. R1). (Page 16)

[Figure]

Fig. R1 Flow chart of the cloud detection algorithm.

The flow chart serves as a summary of the previously described algorithmic process. Figures 8 and 9 then present the corresponding step-by-step results and final determination for a specific case, which directly support the flow chart. Therefore, we believe its current placement is more appropriate.

*Q3. In Section 2.1, it is recommended that authors present a figure or table summarizing the instrument information, its surrounding environment (e.g., location, elevation), and typical sky conditions.*

**Response:** According to your suggestions, we added a table to Section 2.

| Measurement specifications | |
|---|---|
| Temperature range: | -50~400 ℃ |
| Spectral range: | 9.6~11.5 μm |
| Measurement uncertainty: | ±0.5 ℃ plus 0.7 % of the temperature difference between measured target and instrument |
| Long-term stability: | Better than 0.01 % of the absolute measured temperature per month |
| **Deployment environment** | |
| Location: | A rooftop platform at Tibet University's Najin campus |
| Elevation: | 3650 meters |

Table R1: The measurement parameters and deployment environment of the infrared radiometer.

*Q4. Thresholds used in the section 3.2 and section 3.4 need to be clarified, such as 150% of the maximum and the lowest 5% of IRBT values. Authors may provide a brief discussion on how these thresholds were determined.*

**Response:** Thank you for your comment. In the extraction of the clear-sky IRBT diurnal cycle, the wet season window is doubled to capture more clear-sky data due to prolonged cloud cover. The temporal resolution of the clear-sky IRBT diurnal cycle is 20 minutes during the wet season and 10 minutes otherwise, compared to the 2-second observed IRBT resolution. Each window thus contains 600 (wet season) or 300 (other seasons) data points, ensuring statistical confidence. To mitigate cloud contamination, the lowest 5% of IRBT values are averaged as the diurnal cycle value for the target time—using 30 values in the wet season and 15 otherwise. Here the 5% setting is an empirical value, which cannot be justified under the current conditions without other supplementary observations, but works well in practice. Corresponding discussion has been added in the paper. **(**Line 188-191, Section 3.2, Page 11**)**

Following your suggestion, we reassessed the sensitivity of the results to the choice of thresholds. For the spectral test, we adjusted the threshold settings to account for seasonal variations. Due to diurnal temperature variations in different season, the clear-sky IRBT diurnal cycle exhibits seasonal differences. For example, the average maximum IRBT of the clear-sky diurnal cycle (Max-IRBT_DC) in June and November 2021 was 5.86 and 10.6 °C, respectively. Using a uniform 150% increase would result in a high threshold in winter, potentially leading to misjudgments of thin high clouds. Therefore, in the algorithm, when the Max-IRBT_DC exceeds 10°C, the threshold is automatically set to 15°C; otherwise, the threshold is set to 150% of Max-IRBT_DC.

The corresponding revisions have been incorporated into the manuscript. **(**Line 210-213 Section 3.2, Page 12**) (**Line 277-279, Section 3.4, Page 17**)**

[Figure]

Fig. R2 The normalized infrared brightness temperatures (top) and the corresponding calculated standard deviations (bottom) in June 2021. Red reference lines represent the three thresholds - 0.2, 0.25 and 0.3.

For the temporal test, we performed a sensitivity analysis with thresholds of 0.2, 0.25, and 0.3. The results show that during the wet season, the choice of these thresholds has little effect on detection outcomes, as clouds typically exhibit much larger standard deviations (Figs. R2–R3). In winter, the threshold setting may affect the classification in a small fraction of cases (Fig. R4). We further analyzed specific cases, such as November 3, 2021, shown in Fig. R5. Although the IRBT indicates clear skies (top figure), the standard deviation increased to between 0.2 and 0.3 from 4 to 8 UTC (bottom figure). However, without complementary cloud observations, we cannot determine with certainty whether the small fluctuations in standard deviation around 0.2–0.3 were due to clouds or other factors. Based on this sensitivity analysis, we adopt a threshold of 0.3 for cloud detection.

The result of sensitivity analysis has been added in the manuscript. **(Line 245-253, Section 3.3, Page 15)**

[Figure]

Fig. R3 The normalized infrared brightness temperatures (top) and the corresponding calculated standard deviations (bottom) in July 2021. Red reference lines represent the three thresholds - 0.2, 0.25 and 0.3.

[Figure]

Fig. R4 The normalized infrared brightness temperatures (top) and the corresponding calculated standard deviations (bottom) in November 2021. Red reference lines represent the three thresholds - 0.2, 0.25 and 0.3.

[Figure]

Fig. R5 The normalized infrared brightness temperatures (top) and the corresponding calculated standard deviations (bottom) on November 03, 2021. Red reference lines represent the three thresholds - 0.2, 0.25 and 0.3.

*Q5. Some errors should be corrected:*

*Line 58: ' too weak to reliably distinguish' -> 'too weak to be reliably distinguished'*

*Line 62: 'that combines' -> 'that combined'*

*Line 248: 'and are shown' -> 'are shown'*

**Response:** Thank you for your reminding. The above grammatical errors have been corrected.

**References**

Huo, J., Bi, Y. H., Liu, B., Han, C. Z., and Duan, M. Z.: A dual-frequency cloud radar for observations of precipitation and cloud in Tibet: Description and preliminary measurements, Remote Sensing, 13, 4685, https://doi.org/10.3390/rs13224685, 2021.

Luo, J., Pan, Y., Su, D., Zhong, J., Wu, L., Zhao, W., Hu, X., Qi, Z., Lu, D., and Wang, Y.: Innovative cloud quantification: deep learning classification and finite-sector clustering for ground-based all-sky imaging, Atmos. Meas. Tech., 17, 3765–3781, https://doi.org/10.5194/amt-17-3765-2024, 2024.

Liu, B., Huo, J., Lyu, D., and Wang, X.: Assessment of FY-4A and Himawari-8 cloud top height retrieval through comparison with ground-based millimeter radar at sites in Tibet and Beijing, Adv. Atmos. Sci., 38, 1334−1350, Song, X. Q., Zhai, X. C., Liu, L. P., and Wu, S. H.: Lidar and ceilometer observations and comparisons of atmospheric cloud structure at Nagqu of Tibetan Plateau in 2014 Summer, Atmosphere, 8, 9, https://doi.org/10.3390/atmos8010009, 2017.

Song, X. Q., Zhai, X. C., Liu, L. P., and Wu, S. H.: Lidar and ceilometer observations and comparisons of atmospheric cloud structure at Nagqu of Tibetan Plateau in 2014 Summer, Atmosphere, 8, 9, https://doi.org/10.3390/atmos8010009, 2017.

Yan, Y. F., and Liu,Y. M.: Vertical structures of convective and stratiform clouds in boreal summer over the Tibetan Plateau and its neighboring regions, Adv. Atmos. Sci., 36,1089−1102, https://doi.org/10.1007/s00376-019-8229-4, 2019.

Yi, M. J.: Differences in cloud vertical structures between the Tibetan Platea and Eastern China Plains during rainy season as measured by CloudSat/CALIPSO, Advances in Meteorology, 6292930, https://doi.org/10.1155/2019/6292930,2019.

Wang, Y. J., Zeng, X. B., Xu, X. D., Welty, J., Lenschow, D. H., Zhou, M. Y., and Zhao, Y.: Why are there more summer afternoon low clouds over the Tibetan Plateau compared to eastern China? Geophys. Res. Lett., 47, e2020GL089665, https://doi.org/10.1029/2020gl089665, 2020.

Zhao, W., Wang, Y., Bi, Y., Wu, X., Tian, Y., 2, Wu, L., Luo, J., Hu, X., Qi, Z., Li, J., Pan, Y., and Lyu, D.: Unveiling cloud vertical structures over the interior Tibetan Plateau through anomaly detection in synergetic lidar and radar observations, Adv. Atmos. Sci., 41(12), 2381−2398, https://doi.org/10.1007/s00376-024-3221-z, 2024.

---

## Author Comment (AC3)

Dear Reviewer:

We sincerely thank the reviewer for the valuable and constructive comments on our manuscript "An autonomous cloud detection algorithm using single ground-based infrared radiometer for the Tibetan Plateau" (ID: EGUSPHERE-2025-2876). We have carefully revised the manuscript accordingly. Below, we provide point-by-point responses (reviewer's comments in italic, our responses in normal font).

*Q1. My main concern is the selection of the several threshold values. These thresholds (e.g., 150% of clear-sky maximum for spectral test, SD > 0.3 for temporal test) are critical to the algorithm's performance, yet their derivation appears empirical. I suggest the authors conduct a sensitivity analysis to show how performance changes with threshold variation and whether optimal thresholds are season-dependent.*

**Response:** Following your suggestion, we reassessed the sensitivity of the results to the choice of thresholds. For the spectral test, we adjusted the threshold settings to account for seasonal variations. Due to diurnal temperature variations in different season, the clear-sky IRBT diurnal cycle exhibits seasonal differences. For example, the average maximum IRBT of the clear-sky diurnal cycle (Max-IRBT_DC) in June and November 2021 was 5.86 and 10.6 °C, respectively. Using a uniform 150% increase would result in a high threshold in winter, potentially leading to misjudgments of thin high clouds. Therefore, in the algorithm, when the Max-IRBT_DC exceeds 10°C, the threshold is automatically set to 15°C; otherwise, the threshold is set to 150% of Max-IRBT_DC.

The corresponding revisions have been incorporated into the manuscript. **(**Line 210-213 Section 3.2, Page 12) **(**Line 277-279, Section 3.4, Page 17)

For the temporal test, we performed a sensitivity analysis with thresholds of 0.2, 0.25, and 0.3. The results show that during the wet season, the choice of these thresholds has little effect on detection outcomes, as clouds typically exhibit much larger standard deviations (Figs. R1–R2). In winter, the threshold setting may affect the classification in a small fraction of cases (Fig. R3). We further analyzed specific cases, such as November 3, 2021, shown in Fig. R4. Although the IRBT indicates clear skies (top

figure), the standard deviation increased to between 0.2 and 0.3 from 4 to 8 UTC (bottom figure). However, without complementary cloud observations, we cannot determine with certainty whether the small fluctuations in standard deviation around 0.2–0.3 were due to clouds or other factors. Based on this sensitivity analysis, we adopt a threshold of 0.3 for cloud detection.

The result of sensitivity analysis has been added in the manuscript. **(Line 245-253, Section 3.3, Page 15)**

[Figure]

Fig. R1 The normalized infrared brightness temperatures (top) and the corresponding calculated standard deviations (bottom) in June 2021. Red reference lines represent the three thresholds, 0.2, 0.25, and 0.3.

[Figure]

Fig. R2 The normalized infrared brightness temperatures (top) and the corresponding calculated standard deviations (bottom) in July 2021. Red reference lines represent the three thresholds, 0.2, 0.25, and 0.3.

[Figure]

Fig. R3 The normalized infrared brightness temperatures (top) and the corresponding calculated standard deviations (bottom) in November 2021. Red reference lines represent the three thresholds, 0.2, 0.25, and 0.3.

[Figure]

Fig. R4 The normalized infrared brightness temperatures (top) and the corresponding calculated standard deviations (bottom) on November 03, 2021. Red reference lines represent the three thresholds, 0.2, 0.25, and 0.3.

*Q2. I wonder what is the temporal resolution of the detection? Is it a minute basis or daily basis? The authors mention using a sliding window but also compare the BT with the diurnal cycle, so it is not clear whether cloud is detected for every measurement or over the window, or over the course of one day.*

**Response:** The temporal resolution of the detection results is 2 seconds, consistent with the resolution of the observed IRBT data. In the spectral test, the first step is to extract the clear-sky IRBT diurnal cycle, which is a statistical feature with a resolution of 20 minutes in the wet season and 10 minutes in other seasons. This diurnal cycle serves only as a reference for comparison with the observed IRBT data and does not alter the resolution of cloud detection itself.

We acknowledge that the description of temporal resolution in the original manuscript was not sufficiently clear. Based on your comment, we have revised the relevant section to explicitly state: "The temporal resolution of the detection result is the same as that of the observed IRBT data, which is 2 seconds." (Lines 213–215, Section 3.2, Page 12). Other corresponding revisions have been incorporated into the manuscript. **(**Line 87-90, Section 2.1, Page 4-5) **(**Line 176-179, Section 3.2, Page 10)

*Q3. The dust normalization method is innovative, but its assumptions may not always hold—especially during days with persistent cloud cover, when the "daily minimum IRBT" may not represent a dust-free baseline. I suggest the authors separately examine the effect of normalization under strong dust contaminated days. The cloud recognition results should also be evaluated separately for dusty and dust-free days.*

**Response:** Thank you for your valuable comments. As you pointed out, the developed normalization method is unable to fully eliminate the contamination in observational data caused by intense short-term dust deposition. It is worth noting, however, that such events have become increasingly rare in Lhasa. Previous studies reported a significant decreasing trend in dust days since the 1950s due to climate change and ecological improvements, with the annual average declining to 5.2 days in the 1990s and further to 2.7 days between 2000 and 2010 (Zhang et al, 2002; Xu et al. 2007; "The Xizang Meteorological", 2014).

The normalization method developed in this study is primarily intended to mitigate the influence of long-term accumulated dust deposition. Owing to the absence of supplementary observations during the study period, detailed analyses of individual dust events cannot be provided in this study. A clarification regarding the scope of applicability of the method has been added to the methodology section. **(**Line 149-156, Section 3.1, Page 8-9)

*Q4. While radiosonde data are used for validation, the temporal (twice daily) and spatial (4 km offset) mismatch between radiosonde and radiometer observations is significant. This likely underrepresents actual performance. Is it possible to use collocated higher temporal resolution instruments (e.g., ceilometers, cloud radar, or even sky cameras) to validate the results? Or at least some quantitative discussion of potential errors caused by the mismatch should be provided.*

**Response:** Thank you for your suggestion. During the instrument observation period, there were no other simultaneous cloud observations operating at Tibet University or its adjacent areas. Relative humidity data from radiosondes is the only available observation. Therefore, we could only utilize radiosonde data to evaluate the algorithm.

In designing the validation scheme, we considered different physical principles and the temporal inconsistencies between the cloud detections from sounding data and the infrared radiometer. Corresponding constraints and settings were applied to mitigate these discrepancies (Line 319-334, Section 4, Page 20-21).

Furthermore, we provided a detailed analysis of the three main sources causing comparison differences:

(1) spatial mismatch (due to different instrument locations and wind-induced drift of the radiosonde), (2) temporal misalignment (caused by uncertainties in the launch time and ascent rate of the radiosonde), and (3) threshold uncertainty in the radiosonde retrieval algorithm (Line 350-362, Section 4, Page 22; Line 391-394, Section 5, Page 24).

However, owing to the lack of additional meteorological observations, it was not possible to determine which specific factor or combination of factors contributed to the discrepancies, nor to quantitatively assess the impact of each factor on the validation results.

A new dual-wavelength millimeter-wave cloud radar and infrared imager (MWII) has been deployed at the Yangbajing Whole Atmosphere Observatory. This system integrates active radar, providing detailed information on cloud vertical structure and optical properties. Future work will utilize accumulated radar data to conduct more comprehensive evaluation and refinement of the integrated algorithm (Line 407-415,

Section 5, Page 24-25).

*Q5. I also suggest the authors discuss the performance of the algorithm for different cloud types and heights. This, combined with comment 2, might explain part of the poorer performance for the winter season.*

**Response:** Thank you for your suggestion. During the observation period, no other cloud measurements were available at Tibet University or its surrounding areas. The relative humidity from radiosondes was the only supplementary dataset. However, due to retrieval uncertainties and spatial discrepancies, radiosonde data alone cannot reliably identify cloud types or determine cloud (top/base) heights. The lack of complementary cloud observations therefore precludes a detailed evaluation of the algorithm's performance for specific cloud types in this study.

Based on the physical principles underlying the algorithm, we also recognize its potential limitations for certain cloud types. In the spectral test, the relatively high threshold may inevitably cause misclassification—for example, optically thin clouds during nighttime can be erroneously identified as clear skies (Lines 217–218, Section 3.2, Pages 12–13). In the temporal test, clouds with small temporal variability, such as stratus, may likewise be misclassified as clear due to their stable signals (Lines 263–264, Section 3.3, Page 16). The integration of spectral and temporal tests enhances the overall robustness of the algorithm; however, challenges remain under rapidly evolving conditions such as fog or blowing dust, where misclassification is still likely.

Future work will incorporate comprehensive observational datasets to enable a more thorough evaluation and refinement of the integrated algorithm, thereby enhancing its accuracy and applicability under diverse atmospheric conditions. This discussion has been added to Section 5 (Lines 395–401, Page 24).

*Q6. The flow chart of Figure 7 is very important. However, this figure is too simplified lacking detailed information. For example, the normalization strategy and specific thresholds should be added.*

**Response:** According to your suggestion, we revised the flow chart, which involves

more details in normalization and threshold used (Fig. R5).

[Figure]

Fig.R5 Flow chart of the cloud detection algorithm.

*Q7. While prior IRBT-based cloud detection work is cited, the manuscript could more explicitly compare its results with those of similar autonomous algorithms in other regions to highlight relative strengths and weaknesses.*

**Response:** Thank you for your suggestion. The cited related studies in the manuscript all employed comprehensive observational data to develop their algorithms. For instance, the algorithm in Ahn et al. (2015) also utilizes both the spectral and the temporal characteristics of the clouds captured by the infrared radiance from the radiometer. The implementation of this algorithm requires high-temporal-resolution surface observational data (surface air temperature and water vapor pressure at 2 m height), long-term hourly vertical profiles of temperature and humidity with 5 km spatial resolution, and ceilometers or other accurate cloud observation instruments installed adjacent to the radiometer. Based on these observations, dynamic detection thresholds are established to achieve more accurate results.

In contrast, the algorithm proposed in this study relies solely on infrared

radiometer data, making it suitable for cloud observations in remote regions where comprehensive atmospheric measurements are unavailable. Because of the lack of supporting observations, a direct comparison between our algorithm and that developed by Ahn et al. (2015) could not be conducted at present. In the future, we will conduct experimental field observations that can provide necessary observations of both atmospheric states and cloud properties, enabling a detailed comparison of the strengths and weaknesses of the two algorithms, and thereby contributing to further improvements in our method. The relevant prospects regarding this comparison have been added to the last section (Lines 404–406, Section 5, Page 24).

**References**

Zhang, H., Tang, X.: Variation Trends of Climate and Dust Days in Lhasa and Their Relationship (in Chinese), Tibet Science and Technology, 2002(12):51-52. https://doi.org/10.3969/j.issn.1004-3403.2002.12.023, 2002.

Xu, J., Hou, S., Qin, D., Kang, S., Ren, J., and Ming, J.: Dust storm activity over the Tibetan Plateau recorded by a shallow ice core from the north slope of Mt. Qomolangma (Everest), Tibet-Himal region, Geophys. Res. Lett., 34, L17504, https://doi.org/10.1029/2007GL030853, 2007.

The Xizang Meteorological Bureau: The sand-lifting weather in Lhasa has been decreasing year by year. (2014, January 10). Xinhua News Agency. https://www.chinadaily.com.cn/dfpd/xz/2014-01/10/content_17227938.htm